# Decreased CSTB, RAGE, and Axl Receptor Are Associated with Zika Infection in the Human Placenta

**DOI:** 10.3390/cells11223627

**Published:** 2022-11-16

**Authors:** Gabriel Borges-Vélez, Juan A. Arroyo, Yadira M. Cantres-Rosario, Ana Rodriguez de Jesus, Abiel Roche-Lima, Julio Rosado-Philippi, Lester J. Rosario-Rodríguez, María S. Correa-Rivas, Maribel Campos-Rivera, Loyda M. Meléndez

**Affiliations:** 1Department of Microbiology and Medical Zoology, School of Medicine, University of Puerto Rico Medical Sciences Campus, San Juan, PR 00936, USA; 2Department of Cell Biology and Physiology, College of Life Sciences, Brigham Young University, Provo, UT 84602, USA; 3Comprehensive Cancer Center, University of Puerto Rico, San Juan, PR 00936, USA; 4Center for Collaborative Research in Health Disparities, University of Puerto Rico Medical Sciences Campus, San Juan, PR 00936, USA; 5Department of Pathology and Laboratory Medicine, School of Medicine, University of Puerto Rico Medical Sciences Campus, San Juan, PR 00936, USA; 6School of Dental Medicine, University of Puerto Rico Medical Sciences Campus, San Juan, PR 00936, USA

**Keywords:** placenta, trophoblast, Hofbauer cells (HC), Zika virus (ZIKV), tandem mass tagging (TMT)

## Abstract

Zika virus (ZIKV) compromises placental integrity, infecting the fetus. However, the mechanisms associated with ZIKV penetration into the placenta leading to fetal infection are unknown. Cystatin B (CSTB), the receptor for advanced glycation end products (RAGE), and tyrosine-protein kinase receptor UFO (AXL) have been implicated in ZIKV infection and inflammation. This work investigates CSTB, RAGE, and AXL receptor expression and activation pathways in ZIKV-infected placental tissues at term. The hypothesis is that there is overexpression of CSTB and increased inflammation affecting RAGE and AXL receptor expression in ZIKV-infected placentas. Pathological analyses of 22 placentas were performed to determine changes caused by ZIKV infection. Quantitative proteomics, immunofluorescence, and western blot were performed to analyze proteins and pathways affected by ZIKV infection in frozen placentas. The pathological analysis confirmed decreased size of capillaries, hyperplasia of Hofbauer cells, disruption in the trophoblast layer, cell agglutination, and ZIKV localization to the trophoblast layer. In addition, there was a significant decrease in CSTB, RAGE, and AXL expression and upregulation of caspase 1, tubulin beta, and heat shock protein 27. Modulation of these proteins and activation of inflammasome and pyroptosis pathways suggest targets for modulation of ZIKV infection in the placenta.

## 1. Introduction

The Zika virus (ZIKV) is a mosquito-borne flavivirus with high global relevance [1]. This virus is known to cause fever, rash, joint pain, and conjunctivitis [2]. Viral infection can occur in several ways, including mosquito bites, sexual contact, and blood transfusion. Importantly, ZIKV can be transmitted vertically from a pregnant mother to the developing fetus, causing fetal abnormalities known as congenital Zika virus syndrome (CZS) [3]. CZS includes microcephaly, hearing loss, intracranial calcifications, ocular dysfunction, and seizures in newborns [4,5,6]. ZIKV can be attracted to the placenta via direct and contiguous infection of the cell layers, virion transit through a breach, or cell-associated transport [5,6,7]. This virus evades innate human defenses and compromises the placental integrity of the maternal–fetal barriers [6,7]. However, the molecular mechanisms associated with ZIKV penetration into the placenta leading to fetal infection are unknown.

The main structure of the placenta is the chorionic villi which act as a continuous selective barrier that provides gas exchange, nutrient uptake, waste elimination, thermoregulation, hormones, and immunity. This barrier is composed of different trophoblast cell layers. The trophoblast cells mature into syncytiotrophoblasts, which fuse to form the syncytium, the outer layer of the chorionic villi. The cytotrophoblasts are precursors to the syncytiotrophoblasts and the extravillous trophoblasts that anchor the placenta to the endometrium [8]. The syncytiotrophoblast external layer is less permissive to ZIKV infection than the cytotrophoblast layer. This layer is more profound in a healthy placenta and should not be exposed to infection [9]. Placental Hofbauer cells (HCs) are fetal–placental macrophages situated in the intervillous space close to fetal capillaries and provide a conduit for the vertical transmission of some viruses, including ZIKV. Multiple ZIKV strains, including the Puerto Rico strain, successfully infect primary cultures of HCs obtained from term human placentas [4,5]. Due to the proximity of chorionic villi to the fetal blood supply, simple viral multiplication inside HCs may act as a source of virus transmission through fetal blood. However, the mechanism by which ZIKV is resistant to destruction in the placenta is unknown. In recent studies, the lysosomal protease inhibitor cystatin B (CSTB) was found down-regulated in the proteomes of in-vitro-HIV-infected HCs compared to blood-derived macrophages [10]. This suggested an association with HIV-1 restriction in HCs compared to bloodborne monocyte-derived macrophages (MDM) [10]. Further studies confirmed differences in STAT-1 signaling between HC and MDM [11]. In addition, high levels of CSTB interfered with STAT-1 and IFN antiviral responses in MDM, promoting viral replication [12]. However, the role of CSTB in ZIKV placental infection has not been elucidated.

A known hallmark of ZIKV infection is the development of placental inflammation [13,14]. Placental inflammation is associated with fetal and neonatal health and disease [15]. The receptor for advanced glycation end-products (RAGE) is a member of the immunoglobulin superfamily of cell surface receptors expressed in the placenta [16,17,18]. RAGE activation is implicated in inflammation and cell migration processes and is low under physiological settings but can be upregulated in response to inflammation [19]. In this process, activation of RAGE by advanced glycation end products or by damage-associated molecular patterns leads to increased expression and secretion of a variety of inflammatory-response-related molecules, such as TNF α (tumor necrosis factor α), INF γ (interferon γ), and IL-6 (interleukin-6) [19]. In addition, RAGE activation in the placenta is associated with inflammatory diseases such as preeclampsia and intrauterine growth restriction [18]. Interestingly, RAGE receptor activation has been correlated with the activation of the AXL tyrosine kinase receptor [20]. AXL receptors are broadly expressed, and increased activity of this receptor has been implicated in various cellular signaling pathways, including those that influence survival, growth, migration, invasion, and inflammatory responses [21,22]. In the placenta, increased AXL has been implicated in ZIKV infection and inflammatory-associated obstetrics complications, suggesting its essential role in disease development [23,24,25,26,27]. Activation of the RAGE and AXL receptors in the ZIKV-infected placenta as well as the inflammatory pathways affected by the infection still must be elucidated to better understand virus transmission during pregnancy.

If novel preventative and therapeutic strategies are to be created, the mechanism by which ZIKV affects the placental host immune system must be discovered. This work aims to investigate CSTB, RAGE receptor, and AXL receptor expression in ZIKV-infected placental tissues. This will enable the identification of possible mechanisms activated by ZIKV infection of the placenta. We also performed quantitative proteomics and pathway analyses to elucidate the mechanisms and pathways of ZIKV transplacental transmission.

## 2. Materials and Methods

Placental samples: This study was approved by the University of Puerto Rico Medical Sciences Institutional Review Board (IRB). Placentas were collected with approval from subjects of legal age, 21 years or more. Zika virus infection during pregnancy was determined via RT-PCR performed at the Puerto Rico Department of Health. One set of placenta samples (n = 12) was collected from already-processed/examined placentas stored at the Laboratory of Anatomic Pathology, ASEM, Puerto Rico Medical Center, and paraffin blocks were prepared and labeled with a unique number without patient identifiers. Samples were divided into two groups: positive and negative for ZIKV infection. We randomly chose placental tissue from six ZIKV-infected and six negative control women for this study. Placental pathology sections containing chorionic villi were processed overnight (at 4°) and cut into 4 μm thick sections. The second group of fresh frozen placental tissue samples (n = 10) were obtained from the repository of a pilot project, Dental and Craniofacial Effects of Intrauterine Zika Infection (1R21DE027235-01). Proteins from 5 positive and 5 negative ZIKV-infected placentas were extracted for proteomics and for western blot experiments.

Immunofluorescence: Immunofluorescence (IF) was performed on paraffin-embedded placental sections (n = 12) as previously published by our laboratory [28]. Briefly, slides were deparaffinized, followed by treatment with eBioscience™ IHC Antigen Retrieval Solution—Low pH (1X) (Invitrogen, Waltham, MA, USA). Next, slides were incubated overnight (at 4°) with antibodies for NS1 (ZIKV protein; Arigo Biolaboratories Corp., Hsinchu City, Taiwan), cytokeratin (for trophoblast identification; Fitzgerald, Acton, MA, USA), IBA-1 (for macrophages identification; FUJIFILM Wako Pure Chemical Corporation, Osaka, Japan), CSTB (for cystatin B; Abcam plc., Cambridge, UK), AXL (Abcam plc., Cambridge, UK), or RAGE (Abcam plc., Cambridge, UK). Secondary antibodies used were goat anti-mouse Alexa Fluor™ 488, goat anti-rabbit Alexa Fluor™ 546, and goat anti-guinea pig Alexa Fluor™ 647 (Thermo Fisher Scientific, Whatman, MA, USA). Slides were mounted with VECTASHIELD antifade mounting medium with DAPI (Vector Laboratories Inc., Newark, CA, USA) (for nuclear staining). IF was examined using the Eclipse E400 microscope (Nikon Inc., Melville, NY, USA). Pictures were taken with a DS-Qi2 Monochrome Camera (Nikon Inc., Melville, NY, USA) using the NIS-Elements Imaging Software (Nikon Inc., Melville, NY, USA).

Trophoblast cells and ZIKV infection: The trophoblast cell line JEG-3 (HTB-36™) was purchased from ATCC^®^ and cultured in ATCC-formulated Eagle’s Minimum Essential Medium in 10% fetal bovine serum at 37 °C with 5% CO_2_ in T25 and T75 (Corning, Corning, NY, USA) vented tissue culture treated flasks. Cells were exposed for two hours to ZIKV PRVAB59 virus (MOI 0.1) at 50% cell confluency. After two hours, the media containing the virus was removed, replaced with fresh medium, and incubated for 24 h. At the conclusion of the exposure, total cell lysates were obtained.

Western Blot: Western blot was performed as previously published by our laboratory [29]. Briefly, whole tissue homogenates were obtained from frozen placenta using Tissue PE LB™ (G-Biosciences, St. Louis, MO, USA). Cultured cells were lysed, and proteins were extracted with RIPA Lysis and Extraction Buffer (Thermo Scientific, Whatman, MA, USA). Tissues and cell protein lysates (40 μg; *n* = 5 per group) were centrifuged and dried overnight using a Speed Vac (at 4°). Pellets were rehydrated in sample buffer and water, followed by heat at 95 °C for 5 min. Samples were run in a 4–20% TGX gels (Bio-Rad) and transferred to PVDF membranes. Membranes were blocked with EveryBlot Blocking Buffer (Bio-Rad, Hercules, CA, USA) and incubated overnight (at 4°) with antibodies against CTSB (Abcam plc., Cambridge, UK), AXL (Abcam plc., Cambridge, UK), RAGE (Abcam plc., Cambridge, UK), HSP27 (Abcam plc., Cambridge, UK), Tubulin (TUBB; Abcam plc., Cambridge, UK), CASP1 (Abcam plc., Cambridge, UK), and Vinculin (Abcam plc., Cambridge, UK). Membranes were developed using ChemiDoc XRS+ (Bio-Rad) equipment. Band densities were normalized to Vinculin, and comparison between groups were performed.

Statistical analysis of Western blot data: Data are shown as means ± SE. Differences in CTSB, AXL, RAGE, HSP27, TUBB, and CASP1 protein expression were determined between control and ZIKV-positive placentas. Data were analyzed for outliers using ROUT at Q = 1%. Normality was determined using a Shapiro–Wilk test with alpha = 0.05. Statistical differences were determined using an unpaired t-test with *p* < 0.05.

Proteomics Sample Processing: Proteins were isolated from frozen placentas, and concentration was determined as described in the western blot section. Quantitative proteomics protocols were based on Borges et al. 2021 [28]. A total of 10 placentas were processed, and protein extracts (100 μg) were obtained and separated using SDS-PAGE in a Coomassie stained gel for proteomics processing and quantitation. Proteome bands were cut out, and gel pieces were distained by incubation with 50 mM ammonium bicarbonate/50% acetonitrile solution at 37 °C for 2 to 3 h. Proteins were reduced with dithiothreitol (25 mM DTT in 50 mM ammonium bicarbonate) at 55 °C, alkylated with iodoacetamide (10 mM IAA in 50 mM ammonium bicarbonate) at room temperature in the dark, and digested with trypsin (Promega, Madison, WI, USA) overnight at 37 °C at a trypsin/protein ratio of 1:50. The next day, digested peptides were extracted out of the gel pieces using a mixture of 50% acetonitrile/2.5% formic acid in water. Extracted peptides were dried and stored at –80 °C until TMT labeling.

TMT Labeling: TMT reagents are reconstituted in acetonitrile (41 μL for 0.8 mg) on the day of use. As specified by the manufacturer’s protocol (Thermo Scientific, Wathman, MA, USA), dried digests were reconstituted in 100 mM TEAB (triethyl ammonium bicarbonate), TMT labels were added according to the experimental design followed by one-hour incubation with occasional vortexing and a quenching step of 15 min. Finally, equal amounts of each labeled sample were mixed to generate a final pool later submitted to fractionation.

Fractionation: This method was performed using the Pierce High pH Reversed-Phase Peptide Fractionation Kit (REF 89875) and following the manufacturer’s instructions. Briefly, the column was conditioned twice using 300 μL of acetonitrile and centrifuged at 5000× *g* for 2 min, and the steps were repeated using 0.1% trifluoroacetic acid (TFA). Next, each TMT-labeled pool was reconstituted in 300 μL of 0.1% TFA, loaded onto the column, washed, and then eluted 16 times using a series of elution solutions with different acetonitrile/0.1% triethylamine percentages and centrifugation of 3000× *g* for 2 min, generating 16 fractions for analysis. The flow-through step was stored as suggested in the protocol. In case of peptide loss, these can be analyzed if requested.

LC-MS/MS Analysis: Fractions were reconstituted in 0.1% formic acid in water (Buffer A), and a small portion was transferred to autosampler vials for MS/MS analysis using the Easy-nLC1200 (Thermo Fisher Scientific, Wathman, MA, USA). A PicoChip H354 REPROSIL-Pur C18-AQ 3 μm 120 A (75 μm × 105 mm) chromatographic column (New Objective) was used for peptide separation. The separation was obtained using a gradient of 7–25% of 0.1% of formic acid in acetonitrile (Buffer B) for 102 min, 25–60% of Buffer B for 20 min, and 60–95% Buffer B for 6 min. This resulted in a total gradient time of 128 min at a flow rate of 300 nL/min, with an injection volume of 2 μL per sample. Q-Exactive Plus (Thermo Fisher Scientific, Whatman, MA, USA) operates in positive polarity mode and data-dependent mode. The full scan (MS1) was measured over the range of 375 to 1400 at resolution of 70,000. The MS2 (MS/MS) analysis was configured to select the ten (10) most intense ions (Top10) for HCD fragmentation with a resolution of 35,000. A dynamic exclusion parameter was set for 30 s.

Database Search and Results: Mass spectrometric raw data were analyzed using Proteome Discoverer (PD) software, version 2.5. Files were searched against a human database downloaded using the PD Protein Center tool (tax ID = 9606). The modifications included a dynamic modification for oxidation +15.995 Da (M), a static modification of +57.021 Da (C), and static modifications from the TMT reagents +229.163 Da (Any N Term, K). Channel 126 was marked as the control channel, enabling data normalization against the internal pool. The TMT certificate of analysis (Lot: WD312186) was used to correct for isotopic impurities of reporter ions. A series of filters was applied to the PD result file to use those with the highest confidence level, eliminate keratins, and only consider proteins with two or more protein-unique peptides. These filtering parameters reduced protein hits from 4778 proteins to 2881 proteins. These results were exported to Excel for statistics, bioinformatics, and ingenuity pathway analyses.

Statistics, Bioinformatics, and Ingenuity Pathway Analyses: The bioinformatic analysis was performed for the proteomic datasets associated with Zika Virus (5 Zika (+) vs. 5 Zika (–)). The analysis was performed with the Bioconductor software Limma [30,31]. A total of 2797 proteins were processed prior to the statistical analysis. The statistical analysis performed was a single-channel analysis between cases and controls. The results from the statistical analysis were considered to be proteins differentially abundant between groups based on FC ≥ |1.5| and *p*-value ≤ 0.05. For ingenuity pathway analyses (IPA), we selected pathways based on the most significant fold changes and *p*-values together with protein function related to inflammation, tissue remodeling, and protein–protein interactions with CSTB, RAGE, and AXL. IPA tools were canonical pathways, pathway predictions (using the Molecule Activity Predictor tools (MAP)), and protein–protein interaction networks. Ingenuity pathway analysis (IPA) was generated using their software (IPA^®^) (networks, functional analyses, etc.) (QIAGEN Inc., Hilden, DE, USA, https://www.qiagenbioinformatics.com/products/ingenuity-pathway-analysis (accessed on 17 October 2022)) and used for enrichment pathway proteome analysis.

## 3. Results

Zika Virus Placenta infection: We first confirmed placental infection by staining for ZIKV non-structural protein 1 (NS1). NS1 is a protein necessary for viral replication and infection [32]. Immunofluorescence confirmed the presence of NS1 in the placenta of ZIKV-infected mothers (Figure 1A). Interestingly, this protein was localized mainly in the villi trophoblast of infected placentas. This localization was more prominent in the syncytiotrophoblast layer of the placenta (Figure 1A).

Zika Virus infection associated molecules: Macrophages participate in innate immunity and are present in most tissues, such as the liver, skin, gut, lung, and placenta [33]. In the placenta, macrophage hyperplasia was observed during placental infections [34]. We observed increased macrophage infiltration in the ZIKV-infected placentas compared to controls (Figure 1B and Appendix A). This infiltration was observed in the stroma of the placental villi (Figure 1B and Appendix A). Interestingly, when we compared macrophage infiltration with Iba-1 antibody and the expression of ZIKV NS1 protein in the placenta, we observed no colocalization (Figure 1B and Appendix A). The macrophage hyperplasia was observed in the stroma of the placental villi, while ZIKV infection was observed mainly in the outer syncytiotrophoblast layer of the villi (Figure 1A,B, Appendix A).

Cystatin B (CSTB) is a cysteine protease inhibitor that facilitates HIV infection of placental macrophages [10,11,12,35]. In the ZIKV-infected placenta, we detected increased CSTB expression compared to controls (Figure 1C). This increase seems to be distributed in the syncytiotrophoblast layer and in a few in the villi macrophages (Figure 1C and Appendix A).

We decided to investigate the expression of two receptors participating in innate immune responses. RAGE receptor is associated with increased inflammation in several organs [18]. Placental staining showed increased RAGE expression in the ZIKV-infected placenta compared to controls (Figure 1D). This receptor was localized to the stroma in ZIKV-negative placentas, while in ZIKV-positive samples was localized to the placenta’s trophoblast layer, demonstrating increased expression in the syncytiotrophoblast layer (Figure 1D). Interestingly, there was a very low AXL presence in the stroma of ZIKV-negative placentas and no expression in the syncytiotrophoblast layer. However, in ZIKV-positive placentas, an increased AXL expression was observed mainly in the syncytiotrophoblast layer (Figure 1E). To quantify the expression of CSTB, AXL, and RAGE in the placenta, tissue lysates were used to perform western blots. We determined that in ZIKV-infected placentas, there was a significant decrease in CSTB (2.9-fold; *p* < 0.0003), AXL (2.2-fold; *p* < 0.0002) and RAGE (9.7-fold; *p* < 0.0002) expression as compared to controls (Figure 2A). When we examined trophoblast cells infected in vitro with ZIKV, there was a significant decrease in CSTB (1.4-fold; *p* < 0.008) and an increase in AXL (1.4-fold; *p* < 0.008) and RAGE (1.7-fold; *p* < 0.008) expression compared to controls.

To further investigate the inflammatory pathways activated by ZIKV in the placenta, we conducted a quantitative proteomics analysis of frozen placentas from five ZIKV-positive and negative controls using limma Bioconductor software and IPA analyses. As a result, we found 44 differentially more abundant proteins in ZIKV-positive compared to ZIKV-negative placentas, as illustrated in the volcano plot (Appendix A). The list of proteins is described in Appendix A.

Pathway analyses revealed upregulation of proteins associated with the inflammasome (Figure 3), pyroptosis signaling (Figure 4), and the remodeling of the epithelial adherent junction pathway (see Appendix A). The upregulation of these proteins predicts an activation of the pathways mentioned, which could lead to an impaired inflammatory immune response. The upregulated proteins observed in our pathway analysis are caspase-1 (CASP1), serine/threonine-protein kinase Nek7 (NEK7), ubiquitin-associated protein 2-like (Ub), tubulin alpha-1C chain (TUBA1C), and tubulin beta chain (TUBB).

Protein interaction analysis revealed that phosphofurin acidic cluster sorting protein 1 (PACS1) and CASP1 indirectly interact with CSTB and were upregulated (Figure 5). PACS1 indirectly affects the expression of CSTB, and CSTB indirectly activates CASP1. Poly(rC)-binding protein 1 (PCBP1) and small heat shock protein beta 1 (HSPB1/HSP27), which were upregulated in ZIKV-positive placenta, interact directly with AXL (Figure 5). PCBP1 can bind directly with AXL in protein–protein interactions and, AXL activates HSP27 via phosphorylation cascades. These results suggest that infection of ZIKV in the placenta is causing impaired immune responses.

HSP27, TUBB, and CASP1 were selected for further validation using western blot analyses of ZIKV-positive and negative placental tissues. These proteins were selected based on involvement in the pathway analysis literature involving ZIKV interaction and inflammation (Table 1). We found that HSP27 was increased (3.2-fold; *p* < 0.002) in ZIKV-infected placental tissue compared to the negative control (Figure 6). Interestingly, in contrast to quantitative proteomics results, TUBB and CASP1 protein levels were significantly decreased (2.1-fold; *p* < 0.002 and 3.1-fold; *p* < 0.0002) in the western blots (Figure 6).

## 4. Discussion

The Zika virus (ZIKV) is a flavivirus known to infect the placenta and induce fetal abnormalities [39,40]. Although much progress has been made in researching this virus, the mechanism of placental infection remains to be elucidated. Many placental-oriented ZIKV studies have been performed in cell culture models, but actual ZIKV-infected placental tissues are scarce. Therefore, in our laboratory, we became interested in determining signaling molecules associated with this viral infection in the placenta. Concerning the information about the pregnancies, we recognize that it would be ideal to have the data on medical details and the corresponding children. However, these tissues were obtained from the Puerto Rico Department of Health, and no other information was provided to the investigators. The only information obtained was if they were affected by Zika virus or not. The first step of our study included pathological analysis of our placental tissue. Previous pathology analysis in our laboratory revealed decreased size of capillaries, hyperplasia of Hofbauer cells, disruption in the trophoblast layer, and tissue and cell agglutination [41]. Subsequently, we performed proteomics analyses of nine placentas stored in formalin collected during the Puerto Rico 2016 ZIKV epidemic. Quantitative proteomics and ingenuity pathway analysis revealed that 45 of the deregulated proteins in ZIKV-positive compared to ZIKV-negative placentas were cellular components of the extracellular matrix, and 16 played a role in its structure and organization [28]. Of these, fibrinogen was further validated by immunohistochemistry in 12 additional placenta samples and found a significant increase in ZIKV-infected placentas, indicating that infection promotes the coagulation of placental tissue and restructuration of ECM potentially affecting the integrity of the tissue and facilitating the dissemination of the virus from mother to the fetus.

In the current studies, we used frozen placenta tissue from another local repository of ZIKV-infected placentas. We first confirmed ZIKV infection in tissue using immunofluorescence with an antibody against a ZIKV protein, NS1. NS1 protein was mainly localized to the trophoblast layers of the placental tissue compared to the stromal macrophages or HC. These results demonstrated that ZIKV NS1 protein persists in placental tissue until term and is localized in the trophoblast barrier. This result suggests that NS1 protein could contribute to the pathology observed in placental tissue since it has been confirmed that it activates innate immunity, causing cytokine storms, and is involved in immune evasion by binding to the complement system proteins [42]. As expected from previous proteomics analyses [28], the placenta cytokeratin labeling of the trophoblast cell layers confirmed damage in the villi layer of the trophoblasts. This is important as this layer provides a barrier to the developing fetus and is damaged by ZIKV, allowing the exchange of virus-infected cells between the mother and developing fetus [43]. To better understand the role of inflammatory proteins in placental infection by ZIKV, we decided to perform protein expression studies in placental macrophages and trophoblasts from infected and control samples using immunofluorescence. Placental macrophages (Hofbauer cells) are generally present in the placental chorionic villi from as early as 18 days of gestation up to delivery [44]. Macrophage hyperplasia has been reported in ZIKV-infected placentas [41,45]. We confirmed ZIKV-induced macrophage hyperplasia in the stroma of disease patients’ placental tissue compared to negative controls (IBA-1 staining). Unexpectedly, when we compared ZIKV viral protein to macrophage hyperplasia, infection was observed mainly in the trophoblast layer and not in the macrophages. These results and the disruption of extracellular matrix proteins observed previously [28] suggested that macrophage hyperplasia is not caused by direct infection of ZIKV but by the invasion of cells from the blood. To define the players that could be involved in ZIKV infection in the placenta, we explored the expression of proteins previously associated with other viral infections and innate immunity, namely CSTB, RAGE, and AXL. We first wanted to determine the localization and expression of these molecules within the placenta. While CSTB is known to modulate macrophage HIV infection by affecting IFN responses [12], RAGE and AXL are receptors involved in tissue immune responses through nuclear factor kappa B, tumor necrosis factor, and IFN alpha and beta [16,46,47]. We determined that CSTB, RAGE, and AXL localization were primarily expressed in the trophoblast layer that protects the placenta. Interestingly, expression of these molecules was increased in the villi of infected placentas compared to controls. We performed immunoblotting to confirm protein levels in the placental tissues during infection. Interestingly, we determined that expression of these three proteins was significantly decreased in ZIKV-infected placentas compared to controls. This was unexpected as previous bioinformatic analyses have seen an increase in transcriptomic signatures in ZIKV infection of neural cells [48]. However, protein expression can change in different tissues. Previous reports of CSTB protein have shown that this protein is decreased in the proteome of Hofbauer cells compared to blood-derived macrophages [10]. The expression of this protein increases with viral infection and is related to increased STAT-1 signaling and decreased IFN responses [12]. In our studies, we also observed decreased CSTB and ZIKV infection in Hofbauer cells, suggesting a possible mechanism of ZIKV restriction in the placenta. However, ZIKV infection and CSTB expression were concentrated in the trophoblast villi. RAGE receptor has been found in ZIKV infection of monocytes to mediate transmigration in vitro [49]. We expected RAGE to increase due to Hofbauer hyperplasia, but it decreased significantly. Perhaps this could be a mechanism induced by the virus to evade viral immune response in the ZIKV-infected placenta. AXL receptor has been designated as one of the receptors of entry for ZIKV into the placenta, but recent studies showed that it is not required for entry in mouse models of ZIKV infection [26,27,50,51,52]. In our study, AXL shows a significant decrease in expression in ZIKV-positive placentas compared to controls. Although the decrease in AXL receptor was unexpected, it can also suggest a possible mechanism of viral internalization and immune response evasion since AXL is involved in the activation of inflammation and cell survival.

Proteomics analyses revealed important information about a possible activation of the inflammasome and the pyroptosis signaling pathway. Both these pathways are essential in promoting a robust immune response to clear viral infections. Three essential proteins are upregulated in these pathways and contribute to the possible activation of these pathways. These proteins are caspase-1 (CASP1), serine/threonine-protein kinase Nek7 (NEK7), and ubiquitin-associated protein 2-like (Ub). CASP1 plays a central role in the execution phase of cell apoptosis. ZIKV recruits host deubiquitinase to cleave CASP1 [53]. This could explain why we see increased peptides of CASP1 in quantitative proteomics and a decrease of CASP1 in western blot. CASP1 attenuates ZIKV replication [54], and degradation of this protein would be beneficial to the viral propagation. In contrast, activation of an inflammatory response by NEK7 in placental tissue is ideal for ZIKV propagation since cell death caused by the inflammasome and pyroptosis will activate tissue remodeling pathways that were found upregulated in our previous study [28]. Quantitative proteomics detects peptides of the proteins digested while western blot detects antigens linked to a PVDF membrane following antibody treatment. It is possible that the antibody selected did not bind the peptides found in proteomics. Our results also could suggest that overexpression and degradation of CASP1 and TUBB are occurring in ZIKV-infected placental tissue to escape from immune response.

In our previous study [28], we saw how the acute response and coagulation pathways of extracellular proteins are upregulated in a ZIKV infection of the placenta. In our current results, we observed TUBA1C and TUBB, two proteins associated with the remodeling of the epithelial adherent junctions pathway, upregulated. The placental structure’s integrity depends on tight and adherent junctions to protect the developing fetus [55]. Therefore, upregulations of the tubulin proteins could compensate for the loss of integrity caused by ZIKV infection by activating and impairing the immune response.

Our quantitative proteomics results further validate the involvement of the proteins CSTB and AXL in ZIKV infection of the placenta. PACS1 and CASP1 are indirectly associated with CSTB; expression PACS1 affects CSTB and CSTB indirectly activates CASP1 by inflammatory pathways. The expression of PACS1 is beneficial for ZIKV since it directs the localization of furin, a protease that cleaves ZIKV polyprotein, permitting the formation of viral particles [56].

PACS could also play a role in downregulating MHC-1 in the placenta, thus impairing the immune response. Our results show a discrepancy between immunohistochemistry and western blot of CSTB. Immunohistochemistry shows increased CSTB in the trophoblast layers. In contrast, our western blot of whole placental tissue shows a decrease in CSTB in ZIKV-positive cases. It is important to emphasize that immunohistochemistry is better for determining localization since areas with lower expression of proteins of interest can be masked due to saturation in other areas. With our labeling, we cannot distinguish the different types of trophoblast cells and their maturation states. CSTB accumulation can only be observed in the outer layer of the placenta, the syncytiotrophoblasts layers, but the placenta is composed of different types of cells that are in different maturation stages. The western blot was performed on a mixture of proteins that come from all the different types of cells that compose the placenta. CSTB observed in whole placental tissue was decreased. To address the limitation of having a mixture of different cell types in ex vivo tissue, we decided to use the trophoblast cell line JEG-3 in vitro infected with ZIKV. Western blot results show a decreased expression of CSTB in this trophoblast cell line. This contrasts with our immunohistochemistry results, which show expression in the syncytiotrophoblast. It appears that CSTB expression increases only in mature syncytiotrophoblasts and not in cytotrophoblasts. It is important to emphasize that the JEG-3 in vitro model has the limitation of being a single type of cytotrophoblast cell in contrast to the placental tissue having different stages of maturation and different types of cells working as a system. It is also difficult to contrast results of a full-term placenta that has been exposed to ZIKV infection for more time against a cell line that was exposed to ZIKV for a few hours. Taken together, these results demonstrated that ZIKV infection induced a shift in the expression of these inflammatory molecules in trophoblasts and Hofbauer cells, with a significant decrease in their expression in whole placental tissue, while increasing the expression of AXL and RAGE in the trophoblast cell layers. Future experiments should address CSTB expression in the placental tissue by isolating the different cell types of the placenta in an ex vivo model. The decrease in CSTB in whole placental tissue and trophoblasts could be an alternative tissue defense mechanism for suppressing the expression of CASP1 and the formation of the inflammasome [57]. We emphasize that the placenta is an organ that avoids inflammation because it is detrimental to the developing fetus. In contrast, ZIKV uses the host inflammatory system while avoiding it and promoting persistence of infection.

Furthermore, we found two upregulated proteins directly associated with AXL, poly(rC)-binding protein 1 (PCBP1) and HSPB1/HSP27. PCBP1 in other flaviviruses has been seen to benefit in the accumulation of viral RNA [58], leading to more viral particles. HSP27 upregulation was detected previously in our study [28] and in ZIKV-infected cells [59]. HSP27 is a chaperone that responds to environmental stress and causes actin remodeling, cytoskeleton, and membrane organization. AXL could activate the HSP27 and cause an upregulation due to response to impaired inflammation and the tissue damage caused by ZIKV infection. Knockout of HSP70 has provided a protection strategy against ZIKV infection [60]. Upregulation confirmation of HSP27 suggests that placental tissue is goes through stress that could be induced by ZIKV infection in the placenta. Our results suggest that HSP27 could play an important role directly or indirectly in ZIKV infection of the placenta, as seen in our previous quantitative proteomics results and confirmed in our current quantitative proteomics analyses and validation using western blot. Future experiments should be directed at the knockout or downregulation of HSP27 and how ZIKV infection and its pathology are affected in placental cell lines and models.

Taken together, these results suggest that a decrease in CSTB, RAGE, and AXL facilitates viral evasion of immune response to ensure its persistence in the trophoblast layer of the placenta. Our study provides insight into possible key molecules exploited by ZIKV infection to ensure its persistence in the host. Pathology findings may be a secondary effect of ZIKV deregulation of host proteins used in immune evasion, survival, and proliferation.

## Figures and Tables

**Figure 1 cells-11-03627-f001:**
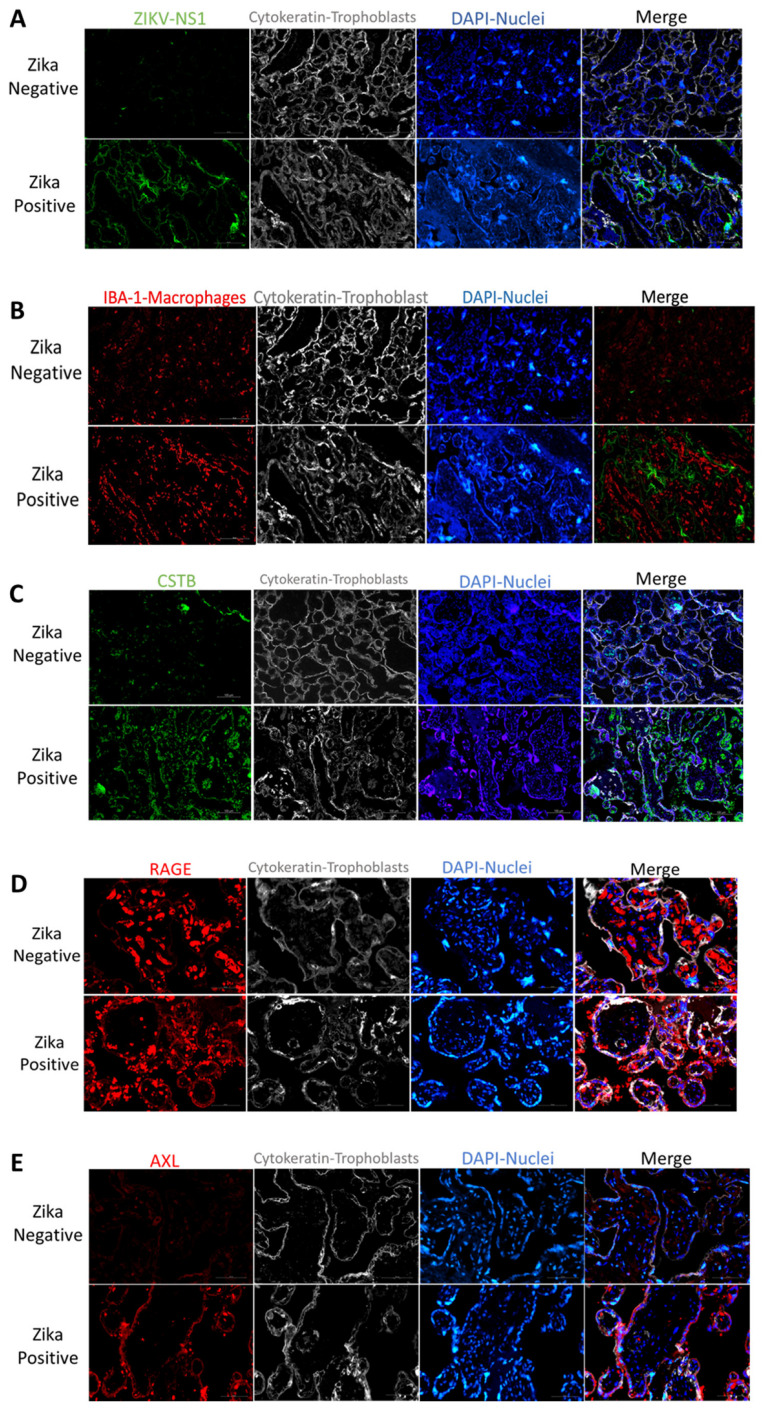
Expression of ZIKV protein and inflammatory proteins in ZIKV-positive and negative placental villi. (**A**) Viral non-structural protein 1 (NS1) labeling of trophoblasts in ZIKV-negative and positive placentas. Representative fluorescence immunohistochemistry of ZIKV-negative placental tissue (upper panel) and ZIKV-infected tissue (lower panel). From left to right, tissue is labeled for anti-NS1 (green), anti-cytokeratin 8 + 18 (white), DAPI for nuclei (blue), and merged figure. Pictures were captured at a magnification of 20×. (**B**) Macrophage labeling of ZIKV-negative and positive placentas. Representative fluorescence immunohistochemistry of ZIKV-negative (upper panel) and positive placentas (lower panel). The tissue is labeled with anti-Iba1 for placental macrophages or Hofbauer cells (red), with anti-cytokeratin 8 + 18 (white) for trophoblast cells, DAPI for nuclei (blue) and anti-NS1 (green). Pictures were captured at a magnification of 20×. (**C**) Cystatin B labeling of ZIKV-negative and positive placentas. Protein labeling of ZIKV-positive (upper panel) and negative placentas (lower panel). Placenta tissue was labeled with anti-cystatin B or CSTB (green), anti-cytokeratin 8 + 18 (white) for trophoblast cells, DAPI for nuclei (blue). Pictures were captured at a magnification of 20×. (**D**) RAGE expression in ZIKV-negative and positive placentas. Representative fluorescence immunohistochemistry of ZIKV-negative (upper panel) and positive placentas (lower panel). Protein is labeled with anti-RAGE (red), anti-cytokeratin 8 + 18 (white) for trophoblast cells, and DAPI for nuclei (blue). Pictures were captured at a magnification of 40×. (**E**) AXL expression in ZIKV-negative and positive placentas. Representative fluorescence immunohistochemistry of ZIKV-negative (upper panel) and positive placenta (lower panel). Protein labeled with anti-AXL (red), anti-cytokeratin 8 + 18 (white) for trophoblast cells, and DAPI for nuclei (Blue). Pictures were captured at a magnification of 40×.

**Figure 2 cells-11-03627-f002:**
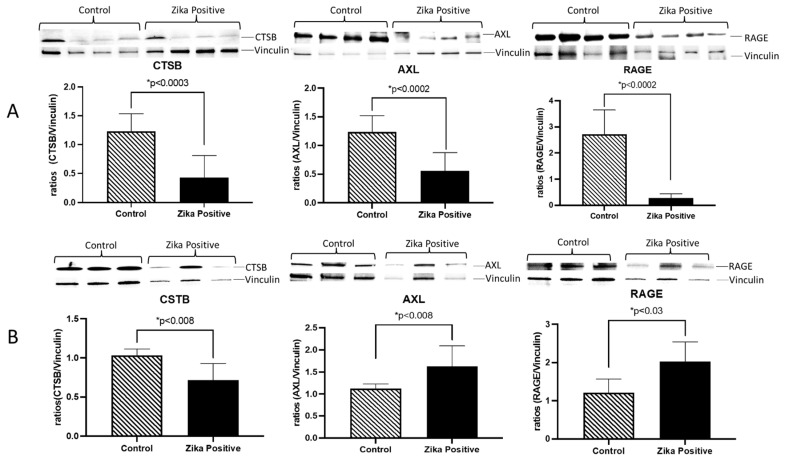
Expression of CSTB, AXL, and RAGE in ZIKV-positive and negative placentas and infected trophoblasts cells. Western blot results from placenta tissue samples showed decreased CSTB, AXL, and RAGE protein levels in the infected placental tissues compared to controls (**A**). Western blot showed decreased CSTB, while AXL and RAGE proteins were increased in infected trophoblast cells compared to controls (**B**). Statistical analysis was performed using Graph Pad 8 from Prism. Statistical differences were determined using an unpaired *t*-test with *p* < 0.05.

**Figure 3 cells-11-03627-f003:**
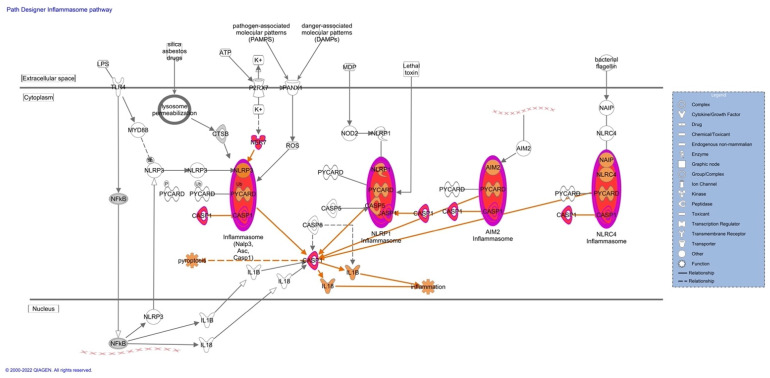
Inflammasome pathways of differentially expressed proteins from ZIKV-positive and negative placentas. Prediction legend: the intensity of red corresponds to higher upregulation of the protein. These proteins are caspase-1 (CASP1), serine/threonine-protein kinase Nek7 (NEK7) and ubiquitin-associated protein 2-like (Ub). The color grey means that the protein was identified but there was no significant increase in expression. Proteins in orange in the diagrams were not detected by our proteomics experiments. The orange means predicted activation of the protein or interaction leading to activation. The protein gene name is used in the diagrams. To identify proteins, refer to Appendix A. Data were analyzed using IPA (QIAGEN Inc., Hilden, Germany, https://www.qiagenbioinformatics.com/products/ingenuitypathway-analysis; accessed 15 September 2022).

**Figure 4 cells-11-03627-f004:**
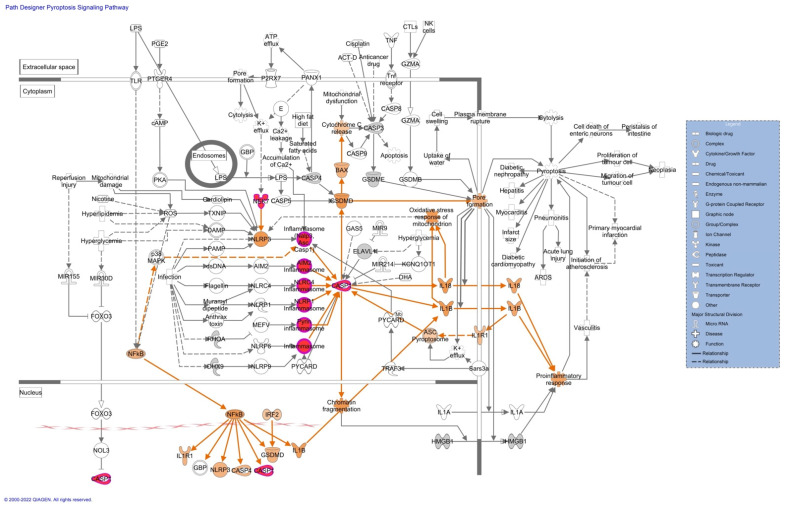
Pyroptosis signaling pathways of differentially expressed proteins from ZIKV-positive and negative placentas. Prediction legend: the intensity of red corresponds to higher upregulation of the protein. These proteins are the caspase-1 (CASP1) inflammasome component and the serine/threonine-protein kinase Nek7 (NEK7). Grey means that the protein was identified but there was no significant increase in expression. Our proteomics experiments did not detect proteins in orange in the diagrams. Orange means predicted activation of the protein or interaction leading to activation. The protein gene name is used in the diagrams. Data were analyzed using of IPA (QIAGEN Inc., https://www.qiagenbioinformatics.com/products/ingenuitypathway-analysis; accessed 15 September 2022).

**Figure 5 cells-11-03627-f005:**
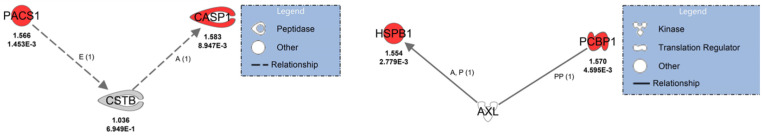
CSTB and AXL protein interactions from differentially expressed proteins from ZIKV-positive and negative placentas are upregulated protein interactions. Prediction legend: the intensity of red corresponds to higher upregulation of the protein. Upregulated proteins are phosphofurin acidic cluster sorting protein 1 (PACS1), caspase 1 (CASP1), poly(rC)-binding protein 1 (PCBP1), and small heat shock protein beta 1 (HSPB1/HSP27). The grey color means that the protein was identified but there was no significant increase in expression. A constant line means direct interaction, and a dashed line is indirect interaction. The letter E means expression; A means activation; P means phosphorylation/dephosphorylation and PP protein–protein binding. Arrowhead means acts on, and a line without an arrow refers to binding only. Data were analyzed using IPA (QIAGEN Inc., https://www.qiagenbioinformatics.com/products/ingenuitypathway-analysis; accessed 15 September 2022).

**Figure 6 cells-11-03627-f006:**
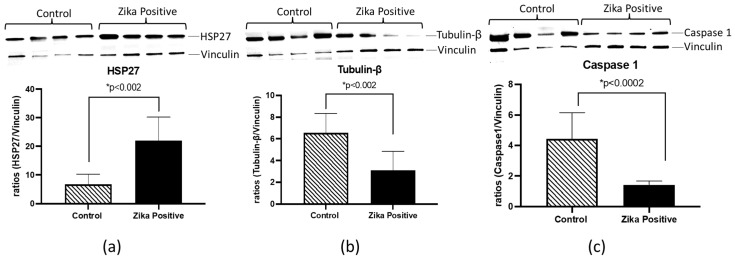
Validation of proteomics data of differentially expressed proteins from ZIKV-positive and negative frozen placentas by western blot. From left to right, we have the densitometry analysis of HSP27, TUBB, and CASP1. Results from relative intensity analysis of protein of interest and representative western blot bands (top of each figure) for this experiment are shown in (**a**–**c**). Relative intensity values for each band were determined using Image Lab Software from Bio-Rad, and ratios were determined using vinculin as a loading control. Statistical analysis was performed using Graph Pad 8 from Prism. Statistical differences were determined using an unpaired *t*-test with *p* < 0.05.

**Table 1 cells-11-03627-t001:** Differentially expressed proteins from ZIKV-positive and negative placenta were selected for validation. Proteins selected for validation of quantitative proteomics. Quantitative proteomics results are included for the selected proteins and brief descriptions.

Protein	Gene Name	Fold-Change	*p*-Value	Function
Caspase-1	CASP1	1.583	0.0089	Thiol protease is involved in various inflammatory processes by proteolytically cleaving other proteins [36].
Small heat shock protein beta 1 (HSP27)	HSPB1	1.554	0.0028	Molecular chaperone probably maintaining denatured proteins in a folding-competent state [37].
Tubulin beta chain	TUBB	1.542	0.0096	A principal constituent of microtubules [38].

## Data Availability

The mass spectrometry proteomics data have been deposited to the ProteomeXchange Consortium via the PRIDE partner repository with the dataset identifier PXD037388 and 10.6019/PXD037388. Data are available via ProteomeXchange with identifier PXD037388.

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
