# Peer review of "Decreased CSTB, RAGE, and Axl Receptor Are Associated with Zika Infection in the Human Placenta"

_cells, 2022, doi:10.3390/cells11223627_

Round 1

Reviewer 1 Report

This study is very well done and describes how Zika virus has an impact on the placenta, which might explain transmission and impact on the fetus.

Two things to address: point out the villi in the images, maybe enlarge an image or add an arrow? It would be great to have a table with the medical details of the placentas and the corresponding children.

Author Response

Reviewer one

Two things to address: point out the villi in the images, maybe enlarge an image or add an arrow? It would be great to have a table with the medical details of the placentas and the corresponding children.

Response: Thank you for this comment. The placental villi are form by the syncytiotrophoblast cells, so staining for Cytokeratin shows where the villi are located by staining the trophoblast layer of the villi. Merging pictures show where the protein of interest is located in comparison to the villi as we stained the trophoblast layer. Concerning the information about the pregnancies, we recognize that it would be ideal to have the data on medical details and the corresponding children, however, these tissues were obtained from Puerto Rico Department of Health and no other information was provided to the investigators. The only information obtained was if they were affected by Zika or not. This sentence was added and highlighted in the first paragraph of the Discussion.

Reviewer 2 Report

1- In the results section, the "overnight" application takes place at room temperature or +4. Should be mentioned.

2- About protein extractions from placental tissue, which part of the placenta? Should be mentioned.

3- In the western blot micrograph should be changed! I would like to see the ladder, housekeeping protein, and researched protein should be shown in the same micrograph. All western blot micrographs should be rearranged accordingly.

Author Response

Reviewer two:

1- In the results section, the "overnight" application takes place at room temperature or +4. Should be mentioned.

Response: Changes were done and highlighted to address this in the manuscript.

2- About protein extractions from placental tissue, which part of the placenta? Should be mentioned.

Response: Concerning the information about the placental tissues, they were obtained  from the Puerto Rico Department of Health and no information was provided about  the location where the placental tissues were obtained. The information we do know is via pathological examination of the placental tissue it was observed that the chorion, including chorionic villi was extracted and seen in the immunohistochemistry.

3- In the western blot micrograph should be changed! I would like to see the ladder, housekeeping protein, and researched protein should be shown in the same micrograph. All western blot micrographs should be rearranged accordingly.

Response: We appreciate this concern. We showed the western as is normally presented in published manuscript. The information requested here can be seen in the original western blot pictures file included in the supplementary data of this submission.  The protein of interest was stripped and the blot was reprobed for the housekeeping protein, vinculin.